# Denoising Single Images by Feature Ensemble Revisited

**DOI:** 10.3390/s22187080

**Published:** 2022-09-19

**Authors:** Masud An Nur Islam Fahim, Nazmus Saqib, Shafkat Khan Siam, Ho Yub Jung

**Affiliations:** Department of Computer Engineering, Chosun University, Gwangju 61452, Korea

**Keywords:** feature ensemble, image denoising, SSIM

## Abstract

Image denoising is still a challenging issue in many computer vision subdomains. Recent studies have shown that significant improvements are possible in a supervised setting. However, a few challenges, such as spatial fidelity and cartoon-like smoothing, remain unresolved or decisively overlooked. Our study proposes a simple yet efficient architecture for the denoising problem that addresses the aforementioned issues. The proposed architecture revisits the concept of modular concatenation instead of long and deeper cascaded connections, to recover a cleaner approximation of the given image. We find that different modules can capture versatile representations, and a concatenated representation creates a richer subspace for low-level image restoration. The proposed architecture’s number of parameters remains smaller than in most of the previous networks and still achieves significant improvements over the current state-of-the-art networks.

## 1. Introduction

Image denoising is a classic problem in the low-level vision domain. A given image X goes through the following mapping to create its noisy counterpart.
Y=X+N

Here, Y is the noisy observation, where N is the additive noise on a clean image X. Denoising is an ill-posed problem, with no direct means to separate the source image and corresponding noise. Hence, researchers follow the best possible approximation of X from Y with corresponding algorithmic strategies.

Typical methods without machine learning involve employing efficient filtering techniques such as NLM [1], BM3D [2], median [3], Weiner [4], etc. Due to their limited generalization capability, additional knowledge-based priors or matrix properties have been integrated into these denoising strategies. However, despite certain improvements with prior-based methods, many concerns remain unresolved, such as holistic fidelity or the choice of priors.

Convolution neural network (CNN) denoising methods later offered an unprecedented improvement over the previous strategies through their customized learning setup. Usually, CNN methods offer better performance through brute force learning [5], tricky training strategy [6], or inverting image properties [7] by various proposals. We observed gradual improvements over the years for denoising solutions. However, these methods with pure brute force mapping sometimes face fidelity issues within challenging noisy images. Furthermore, due to the lack of generalization properties, the methods provide reconstructed images that often result in cartoonized smoothing.

In contrast, the proposed approach rebuilds a previous ensemble-oriented denoising network that can successfully estimate a cleaner image with less cartoon-like smoothing. For the design of the proposed denoising network, we carefully maximized detail restoration by providing a variety of low-level ensemble features while keeping the network relatively shallow to prevent an oversized receptive field and hallucination effects. In summary, our study has the following contributions:We propose a shallow ensemble approach through feature concatenation to create a large array of feature combinations for low-level image recovery.Due to the ensemble of multiple modules, our model successfully returns fine details compared to previous data-driven studies.The parameter space is relatively small compared to the contemporary methods with a computationally fast inference time.Finally, the proposed study shows better performance with a different range of synthetic noise and real noise without the cartoonization and hallucination effect. See Figure 1.

## 2. Related Work

### 2.1. Filtering Based Schemes

Traditional filtering approaches aim for handcrafted filters for noise and image separation. These studies [3,4] utilized low-pass filtering methods to extract the clean images from the noisy images. The iterative filtering approach adopted a progressive reduction for image restoration [9]. Additionally, several methods used nonlocal similar patches for noise reduction based on the similarity between the counterpart patches in the same image. For example, NLM [1] and BM3D [2] assumed a redundancy within patches from a given image for noise reduction. Nonetheless, these methods usually produce flat approximations, as the given image severely degrades the noisy image quality with a heavy noise presence.

### 2.2. Prior Based Schemes

Another group of studies focused on selecting priors for the model, which produce clean images when optimized. These methods reformulated the denoising problem as a maximum a posteriori (MAP)-based optimization problem, where the image prior regulated the performance of the objective function. For example, the studies [10,11] assumed sparsity as the prior for their optimization process. The primary intuition was to represent each patch separately through the help of a function. Xu et al. [12] performed real-world image denoising by proposing a trilateral weighted sparse coding scheme. Other studies [12,13,14,15] focused on rank properties to minimize their objective function. Weighted nuclear norm minimization (WNNM) [12] calculated the nuclear norm through a low-rank matrix approximation for image denoising. Additionally, there are several complex model-based derivations using graph-based regularizers for noise reduction. However, their performance degrades monotonically for noisier areas, and recovering the detailed information is sometime difficult [16,17,18]. Additionally, these methods generally output significantly varying results depending on their prior parameters and the respective target noise levels.

### 2.3. Learning Based Schemes

Due to the availability of paired data and the current success of CNN modules, data-driven schemes have achieved significant improvements in separating clean images from noisy images. Recent CNN studies [19,20,21] utilized the residual connection for estimating the noise removal map before inference. These studies evaluated the clean image without taking any priors regarding the structure or noise. They achieved enhanced performance by using a noncomplex architecture with repeated convolutional, batch normalization, and ReLU activation function blocks. However, these methods can fail to recover some of the detailed texture structure in the presence of heavy noise area.

Trainable nonlinear reaction diffusion (TRND) [22] used a prior in its neural network and extended the nonlinear diffusion algorithm for noise reduction. However, the method suffered from computational complexity as it required a vast number of parameters. Similarly, the nonlocal color net [23] utilized the nonlocal similarity priors for the image denoising operation. Although priors mostly aid the denoising, there are some cases where the adaptation of the priors degrades the denoising performance. Very recently, DEAMNet [8] surpassed the previous state-of-the-art results by using an adaptive consistency prior.

With the success of the DnCNN [19], two similar networks called “Formatting net” and “DiffResNet” were proposed with different loss layers [5]. Later, Bae et al. [24] proposed a residual learning strategy based on the improved performance of a learning algorithm using manifold simplification, providing significantly better performance. After that, Anwar et al. [25] proposed a cascaded CNN architecture with feature attention using a self-ensemble to boost the performance.

A few recent approaches [26,27] followed the blind denoising strategy. CBDNet [27] proposed a blind denoising network consisting of two submodules, noise estimation and noise removal, by incorporating multiple losses. However, their performance was limited by manual intervention requirements and a slightly lower performance on real-world noisy images. In comparison, FFDNet [28] achieved enhanced results by proposing the nonblind Gaussian denoising network. Consequently, RIDNet [5] utilized perceptual loss with $\ell_{2}$ apart from the DnCNN architectures for noise removal and achieved significant success by introducing a single-stage attention denoising architecture from real and synthetic noises. Liu et al. [7] introduced GradNet by revisiting the image gradient theory of neural networks. Recently, several GAN-based approaches [29,30,31,32] were introduced through generating denoised images following either a data augmentation strategy for creating diverse training samples or a strategy based on the distribution of the clean images.

The tendency for a modern denoising machine learning scheme is to use a deeper network with complex training. However, Figure 1 shows some of the hallucination and oversmoothing problems of deeper networks. A hallucination of a windowed building can be observed in the DEAMNet [8] results in Figure 1. We believe that deeper network architecture with its overfitting tendencies is the cause of the hallucination and over-smoothing. We also suspect the PSNR minimization is contributing to oversmoothing. Thus, in this paper, we propose a variety of shallow networks for low-level feature accumulation as well as a network that finds a balance between PSNR and SSIM. We show that multiple feature ensembles from a variety of shallow networks are more appropriate for denoising image problems compared to a single deeper and complex network. The shallow architecture prevents overfitting, and the necessary statistics are obtained from the feature ensemble.

## 3. Methodology

### 3.1. Baseline Supervised Architecture

Recently, supervised model-based denoising methods embedded a similar baseline formation in their proposals [5,33]. In brief, it is possible to compartmentalize the baseline architecture in Figure 2 into three distinct modules: initial feature extraction, large intermediate blocks, and feature reconstruction. Typically, the primary module consists of a single layer that serves the purpose of the initial feature learner or initial noise estimator. For any noisy image $I_{n}$, the representation of the initial block $E_{p}$ is as follows:(1)$$E_{p}=\beta \left(I_{n}\right)$$
where β is the initial convolutional layer for basic feature training. Following that, we see the main restoration part of the given network through the help of an intermediate processor. Typically, this intermediate layer is a very long cascaded connection of the unique feature extractor units. From time to time, we often observe the presence of long residual-dense or residual-attention blocks as the backbone of such setup.

Now, if the intermediate block is M, the cascaded representation of the intermediate processing stage is as follows:(2)$$E_{i}=M_{j}\left(M_{i-1}\left(.....\left(E_{p}\right)..\right)\right)$$
where $M_{i}$ represents the *i*th instance of the learning stage of the intermediate block and $E_{i}$ is the corresponding outcome of the intermediate layer.

The final reconstruction module operates through a residual connection followed by a consecutive final convolution. If R is the final reconstruction stage before the output, then the recovered image $I_{r}$ is a combination of $E_{i}$, $E_{p}$, and $I_{n}$.
(3)$$I_{r}=R\left(E_{i},E_{p},I_{n}\right)=f_{N}\left(I_{n}\right)$$

Here, $f_{N}$ denotes the overall neural network, $I_{n}$ is the noisy input image, and $I_{r}$ is the recovered image. A typical choice of the cost function for this task involves the $\ell_{1}$ or $\ell_{2}$ loss. There are other customized loss functions available, such as weighted-augmentation of different loss functions that integrate spatial properties or relevant regularization [5]. In general, the network is optimized by minimizing the difference from clean images.
(4)$$\zeta \left(\theta \right)=\frac{1}{N}\sum_{i=1}^{N}\left|\right|f_{N}\left(\theta ,I_{n}^{i}\right)-I_{c}^{i}{\left|\right|}_{1}$$

Here, θ is the learnable parameter, $I_{n}^{i}$ is the noisy image, and $I_{c}^{i}$ is the corresponding clean image. Most of the baseline network parameters are placed in the intermediate learning block.

### 3.2. Proposed Architecture

In contrast, we designed our network to allocate more resources to the concatenated learned features. Instead of developing a basic learning block for long cascading connections, we chose variety by proposing various individual feature learning blocks. The proposed network is focused on delivering richer and diverse low-level features. To further reduce complexity, we avoided using an attention operation, which is typically more expensive. More details are provided in Figure 3 and the following subsection.

#### Initial Feature Block

Three consecutive convolution layers are used to extract the initial features for the network. The layers are equal in depth, but their kernel sizes were in descending order. The input image goes through the 5×5 convolution operation at the first layer, followed by a 3×3 convolution, and ends at the 1×1 pixelwise operation. A larger kernel size makes use of a larger neighborhood of input features and estimates the representations on larger receptive fields. By limiting the kernel size and the number of layers, the network learns to focus on the smaller receptive fields and disregards the broader view, which we argue to be less meaningful in low-level vision tasks such as denoising. Therefore, the purpose of the primary layer is to project the representation for the denoising features from a smaller receptive field into individual responses which can be further diversified in the next four block modules in Figure 3.

### 3.3. Four Modules for Feature Refinement

Before presenting the four modules for feature refinement, we cover the convolution, activation functions, and residual connections used in the modules. Even though the attention mechanism is a common choice to learn richer representations, we can still find a similar or better result without it in this study. Additionally, our selection of residual blocks was for blocks to be no longer than six consecutive connections. The fundamental operations for our modules are introduced below.

**Convolution.** In the internal convolution operation, our choice of kernels varied from 1×1 to 7×7. Due to such a range, our network was naturally focused on both smaller and larger receptive fields.

**Activation functions.** Recent advancements in nonlinear activation functions have shown that better performance is achievable through the interconnected operation of different activation representations that are compacted into a single function. Hence, we chose the SWISH [34] and MISH [35] activation representations in addition to the ReLU operations. As a result, our network learned from diverse representations obtained from various parallel activated functions.

**Residual connections.** It is redundant to mention the efficacy of the residual connections in the vision tasks. In the literature, we can see that the customization of residual connections varies within the task. In the original ResNet paper [36], the authors included batch-normalization between the convolution layer, followed by the ReLU layer. In our study, we used the convolution layers, which were separated by the ReLU layer. This choice of the ReLU sandwich residual connection is prevalent in regression tasks [37].

We focus on the major processing modules below with the description of the utilized blocks. We propose four processing modules that perform the refinement operations on the initial features. The following subsections cover their descriptions and the basic reasoning behind the proposed architecture.

#### 3.3.1. Residual Feature Aggregation Module

In our residual feature aggregation module, we used the aforementioned residual blocks as our underlying design mechanism. In the construction of this module, we took inspiration from the traditional pyramid feature extraction [38] and aggregation, which has been very influential in computer vision. A typical pyramid setup is motivated by the needs for multiscale feature aggregation, which, in essence, utilizes low-frequency information along with high-frequency features. However, the subsequent downsampling process is a lossy operation by nature. To mitigate information loss for low-frequency features, we chose to employ the concurrent residual blocks on the same initial features through three different kernel sizes. Naturally, our kernel choice ranged from 1×1 to 5×5, as seen in Figure 4a. Hence, a larger kernel allowed us to learn the features from a larger area of image, while the 1×1 kernel operation allowed us to maintain the initial receptive field and make use of more high-frequency information. We aggregated the response from all three residual block to learn the overall multiscale impact of the initial features. Finally, a typical 3×3 convolution with standard depth gave us the *n* number of diverse representations from this module. As a result, our model can learn the important multiscale features without going through a pooling operation.

#### 3.3.2. Multiactivation Feature Ensemble

Activation functions are unavoidable components for neural network construction that aid the learning operation by projecting the impactful information to the next layer. Hence, widely different nonlinear functions are available as activation functions in all sorts of neural networks for various purposes. The ReLU is the most widely used activation function, which at heart is a “positive pass” filter. However, in some cases, zero-out negatives and a discontinuity in the gradient are argued to be unhelpful in the optimization process. To address some of its weaknesses, SWISH [34] and MISH [35] were proposed with smooth gradients while maintaining a similar positive-pass shape of the ReLU. A recent experiment [35] showed that these activation functions provided a smoother loss landscape than the ReLU.

Nonetheless, we incorporated all three activation functions, as seen in Figure 4b. SWISH, MISH, and ReLU activation functions were applied to the initial features, followed by a convolution layer. The subsequent responses were concatenated into a single tensor to learn from the integrated representation of varying activation functions. No further kernels and residual blocks were utilized for this module. The initial feature results of these modules were ensembled with the responses of the other three modules, but the multiactivation functions were also integrated into the multiactivated cascaded aggregation module described in Section 3.3.3.

#### 3.3.3. Multiactivated Cascaded Aggregation

In this module, both shallow and relatively deeper layer features were concatenated. Typically, a deep consecutive convolution operation is formulated after the initial feature extraction, and the conventional thinking is to build a deeper network for complex problems. However, we added a single convolution layer feature to complement the deeper layer features because we believed that a shallower interpretation might be more appropriate for low-level vision problems. See Figure 5.

For a single convolution path, a 3×3 kernel size was chosen with the same depth as the initial features. For the deeper path, five consecutive convolution layers with different kernel sizes were used. The activation functions between the layers were ReLUs, however, for both paths, a multiactivation feature ensemble was implemented as described earlier. Both the shallow and deeper responses were concatenated followed by another convolution layer.

#### 3.3.4. Densely Residual Feature Extraction

The densely residual operation has shown great promise in both regression and classification tasks [39]. Dense residual connections are an efficient way to emphasize hierarchical representation. For this reason, we designed a densely residual module to aggregate features for the network. The proposed design in Figure 6 also utilized the concatenation between the final and previous aggregation in support of a total hierarchy concentration. A final convolution was added to combine the three concatenated features from the densely residual layers.

After collecting and concatenating the individual responses from each of the four modules, the responses were merged by the final convolution layer with a dilation rate of 2, see the overall process in Figure 3. This layer’s output contained the most refined representation for the restored image. The restored image was fed into a simple loss function consisting of $\ell_{1}$ and $\ell_{SSIM}$.

### 3.4. Loss Function

We used two typical loss functions, $\ell_{1}$ and $\ell_{SSIM}$, to update the parameter space. The total loss function was a simple addition of the two.
(5)$$\ell_{total}=\ell_{1}+\ell_{SSIM}.$$$\ell_{1}$ measures the distance between the ground truth, clean image and the restored image as shown in next equation.
(6)$$\ell_{1}=\frac{1}{n}\sum_{j=1}^{n}\left|\gamma_{g}-\gamma_{p}\right|.$$

Here, $\gamma_{g}$ is the ground truth clean image and $\gamma_{p}$ is the restored prediction image. The secondary component is the loss function from SSIM, which is another widely used similarity measure for images.
(7)$$\ell_{SSIM}=\frac{1}{n}\sum_{j=1}^{n}1-SSIM\left(\gamma_{g},\gamma_{p}\right)$$

## 4. Experimental Results

This section describes the overall performance of our method on both real and synthetic noisy images.

### 4.1. Network Implementation and Training Set

For the proposed study, we utilized a TensorFlow framework with NVIDIA GPU support. Most of the convolutional layers in our network were 3×3 kernels, apart from the specific cases where 1×1, 5×5, and 7×7 kernels in addition to the 3×3 kernels were used. For the training phase, we used the method from He et al. [40] for the initialization and the Adam optimizer with a learning rate of $10^{-4}$, a typical default in many vision studies.

For the training, the DIV2K dataset was used. To enable diversity in the data flow, the typical rotation, blurring, contrast stretching, and inverse augmentation techniques were implemented. The training images were cropped into smaller patches. The noisy input images were created by perturbing the clean patches by additive white gaussian noise (AWGN) with 15, 25, and 50 standard deviations.

### 4.2. Testing Set

We use the BSD68, Kodak24, and Urban100 datasets for the inference comparison, where clean observations were available and noisy versions were created through the same artificial noise augmentation. The results are summarized in Table 1.

The DND, SIDD, and RN15 datasets were used to evaluate the proposed approach on images with natural noise. A brief description of the real-world noisy image dataset and the evaluation procedures are described below.

**DND**: DND [41] is a real-world image dataset consisting of 50 real-world noisy images. However, near noise-free counterparts are unavailable to the public. The corresponding server provides the PSNR/SSIM results for the uploaded denoised images.**SIDD**: SIDD [42] is another real-world noisy image dataset that provides 320 pairs of noisy images and near noise-free counterparts for training. This dataset follows a similar evaluation process as for the DND dataset.**RN15**: RN15 [26] dataset provides 15 real-world noisy images. Due to the unavailability of the ground truths, we only present the visual result of this dataset.

### 4.3. Denoising on Synthetic Noisy Images

For evaluation purposes, we considered previous state-of-the-art studies within various contexts. The evaluation procedure included two filtering methods, BM3D [2], WNNM [12], and several convolutional networks including DnCNN [19], FFDNet [28], IrCNN [20], ADNet [43], RIDNet [5], VDN [44], and DEAMNet [8].

Table 1 shows the average PSNR/SSIM scores for the quantitative comparison. From the average PSNR and SSIM score, the proposed study surpasses the previous studies with a considerable margin. We adopted three widely used datasets BSD68, Kodak24, and Urban100 with three different AWGN noise levels, 15, 50, and 50. The code for all methods used in this evaluation, including our own source code, is found in Appendix A.

For a visual comparison, Figure 7, Figure 8 and Figure 9 from BSD68, Kodak24, and Urban100 are presented, respectively, with a noise level of 50. Figure 7 shows the “fireman” picture from the BSD68 dataset. The differences in the restoration are shown in detail with a more controlled smoothing. From Figure 8, we see that the proposed approach avoids image cartoonization and preserves details while restoring clean details. The proposed study manages to restore the structural continuity compared to other methods while preserving the appropriate color and contrast of the image. The last visual comparison for the synthetic noisy image is the “Interior” picture from the Urban100 dataset, shown in Figure 9. For a better illustration of the differences, a zoomed image of the interior wall of the place is shown, where the proposed method manages to preserve the brick’s separating lines more clearly. We also provide Figure 10, where multiple images were combined with different intensities of noise with their proposed output.

### 4.4. Denoising on Real-World Noisy Images

The results for real-world noisy image restoration are presented in Table 2. Natural noise removal is challenging because the convoluted noises are not signal independent and vary within the spatial neighborhood.

We chose three real noisy image datasets, the SIDD benchmark [42], the DnD benchmark [41], and RN15 [26], to analyze the generalization capability of our proposed method. For the SIDD and DnD benchmarks, the clean counterpart images are not openly distributed. Hence, the presented PSNR/SSIM in Table 2 was obtained by uploading the results into the corresponding server. For the RN15 dataset, there is no benchmark utility. Table 2 represents the comparative performance for both SIDD and DnD benchmarks. Among the existing methods, VDN [44] and DEAMNet [8] perform well. However, our method achieves a better result among the existing methods for both the real and synthetic noises.

To demonstrate the performance of our method with real images, we also provide some visual comparisons in Figure 11, Figure 12 and Figure 13 on the SIDD, DND, and RN15 datasets, respectively. For a visual comparison on real noisy images, we included the recent VDN [44], RIDNet [5], and DEAMNet [8]. The visual comparison shows that our method tends to avoid cartoonization while effectively removing noise, suppressing artifacts, and preserving object edges. Overall, the qualitative and quantitative comparisons display an effective performance on all fronts.

### 4.5. Computational Complexity

This section provides a comparison of the computational complexity through Table 3. The table represents the average running times for the three different image sizes 256×256, 512×512, and 1024×1024. In addition, we present the parameter counts of the compared methods. Apart from BM3D [2], we report the model-specific computation time. In this comparison, we considered BM3D [2], DnCNN [19], WNNM [12], IrCNN [20], FFDNet [28], AINDNet [33], ADNet [43], VDN [44], RIDNet [5], and DEAMNet [8]. In Table 3, our method’s computation time is only slightly longer than the earlier DnCNN, IrCNN, FFDNet, ADNet, and VDN. In terms of parameter counting, the proposed study is significantly smaller than the recent RIDNet [5], AINDNet [33], VDN [44], and DEAMNet [8].

### 4.6. Ablation Study on Modules

In this section, we provide an ablation study based on the effect on our modules’ correlation. We used four different modules, which work separately and generate various features. These different features cannot be considered separately as clean images. However, if we concatenate them together as the proposed method described, we can obtain a clean image. In Table 4, the modules are the residual feature aggregation block (RFA), multiactivation feature ensemble block (MFE), multiactivated cascaded aggregation block (MCA), and densely residual feature extraction block (DRFE). We removed each module separately and calculated the PSNR and SSIM for three different datasets. Here, we can observe that the PSNR value drops every time a module is removed. For the multiactivation feature ensemble block (MFE), the value of the PSNR drops the most, and for the module multiactivated cascaded aggregation block (MCA), the SSIM value drops the most. In Figure 14, we represent the output of these four modules separately with the ground truth and our proposed method’s output.

## 5. Conclusions

In this paper, the basic strategy for the low-level denoising problem was to gather a variety of low-level features while keeping the interpretation simple by implementing relatively shallow layers. We argued that for low-level vision tasks, the principle of Occam’s razor was more appropriate, and accordingly, we designed a network that focused on gathering a variety of low-level evidence rather than providing a deep explanation of the evidence. Thus, we revisited the feature ensemble approach for the image denoising problem. Our study offered a new model which concatenated different modules for creating large and varying feature maps. To enhance the performance of our network, we utilized different kernel sizes, residual and densely residual connections, and avoided deep unimodule cascaded aggregation. We carefully designed four different modules for our study, where each helped to restore different spatial properties. Finally, we validated our network with natural and synthetic noisy images. Extensive comparisons showed the overall efficiency of the proposed study. We observed that although our SSIM scores were much higher across the board, the PSNR scores were not the best in the comparison. Our model extracted a variety of shallow features from the image; however, for higher PSNR evaluation, a deeper network may be desirable. In future work, we are planning to apply a self-supervised strategy in training procedures using the same ensemble of shallow networks. The different versions of the noisy input images are planned to be used during the denoising self-supervised training.

## Figures and Tables

**Figure 1 sensors-22-07080-f001:**
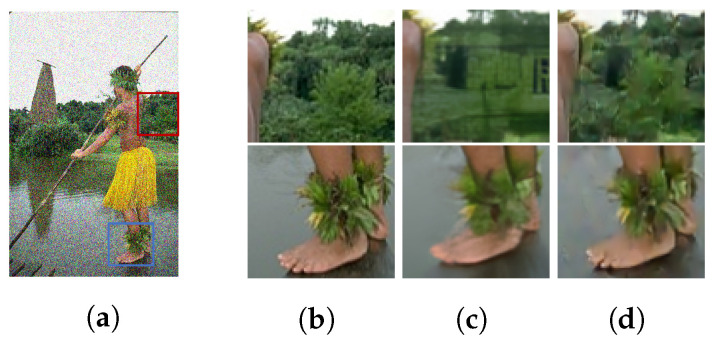
Demonstration of our contribution. An image from the BSD68 dataset with additive white gaussian noise (AWGN) noise with σ = 50. Here, the first column shows the ground truth, followed by the inference from the DEAMNet [8], and the final column shows the proposed result. From a side-by-side comparison, the proposed method can restore the image without the hallucination effects of deeper networks. (**a**) Noisy. (**b**) Ground Truth. (**c**) DEAMNet [8]. (**d**) Proposed.

**Figure 2 sensors-22-07080-f002:**
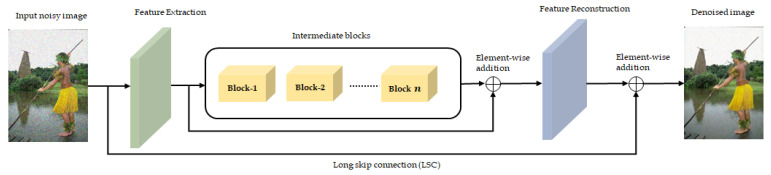
This figure shows the general baseline architecture for the denoising model, which usually consists of the more prolonged feature extraction phase with cascaded modules, which begin right after the initial feature collection and end with the final residual aggregation.

**Figure 3 sensors-22-07080-f003:**
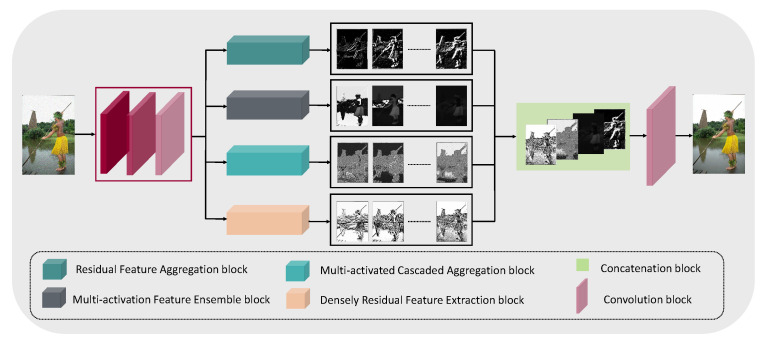
In the above figure, we present the overall diagram of the proposed architecture for image denoising. Our pipeline first extracts the initial feature using consecutive convolution operation, followed by the four modules for feature refinements. These modules are standing upon the customized convolution and residual setup with supportive activation functions. After refinement, we concatenate all the refined feature maps into a single layer, followed by a final dilated convolution to make the inference.

**Figure 4 sensors-22-07080-f004:**
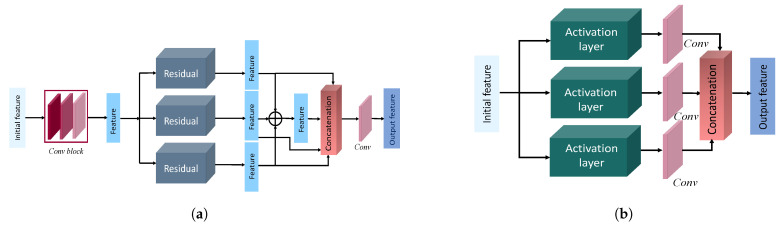
The first two modules in our proposed architecture. The first one is the residual feature aggregation module, and the second one is the multiactivation feature ensemble module. (**a**) Residual feature aggregation module. (**b**) Multiactivation feature ensemble module.

**Figure 5 sensors-22-07080-f005:**
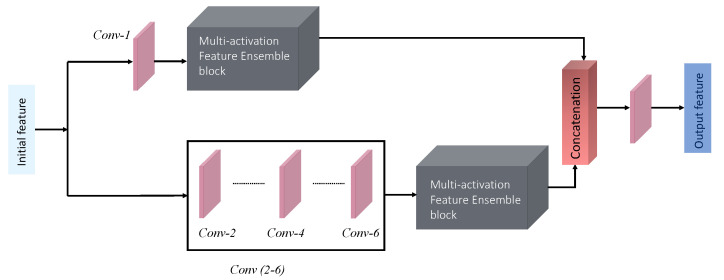
Multiactivated cascaded aggregation module.

**Figure 6 sensors-22-07080-f006:**
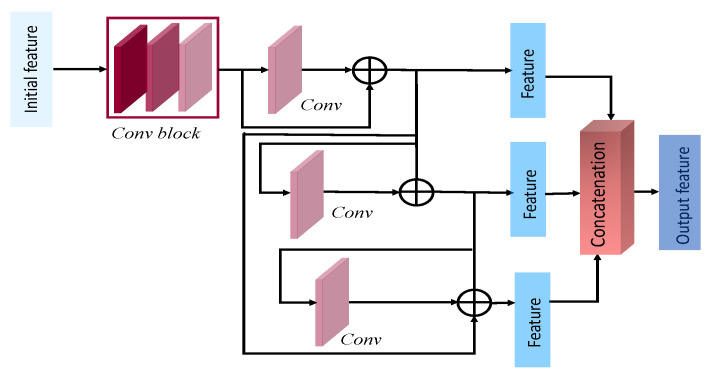
Densely residual feature extraction module.

**Figure 7 sensors-22-07080-f007:**
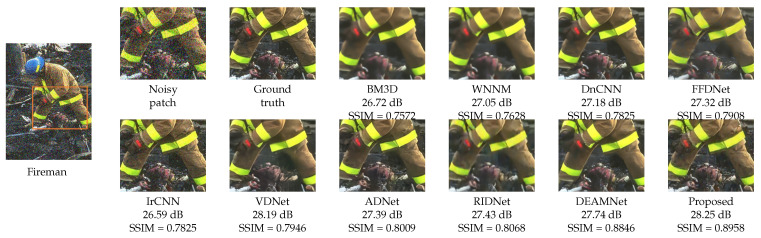
Visual quality comparison with PSNR and SSIM scores for “Fireman” from the BSD68 dataset with AWGN noise level σ=50 (for best view, zooming in is recommended).

**Figure 8 sensors-22-07080-f008:**
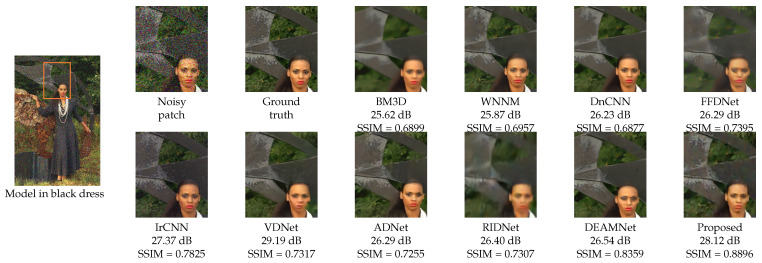
Visual quality comparison with PSNR and SSIM scores for “Model in black dress” from the Kodak24 dataset with AWGN noise level σ=50 (for best view, zooming in is recommended).

**Figure 9 sensors-22-07080-f009:**
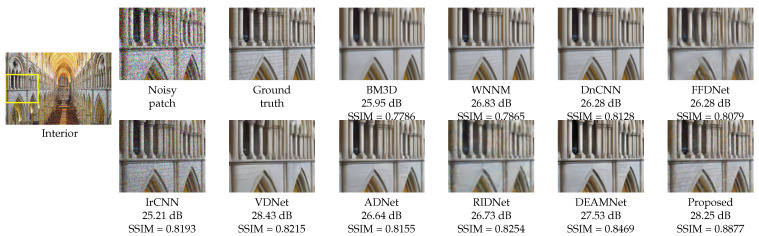
Visual quality comparison with PSNR and SSIM scores for “Interior” from the Urban100 dataset with AWGN noise level σ=50 (for best view, zooming in is recommended).

**Figure 10 sensors-22-07080-f010:**
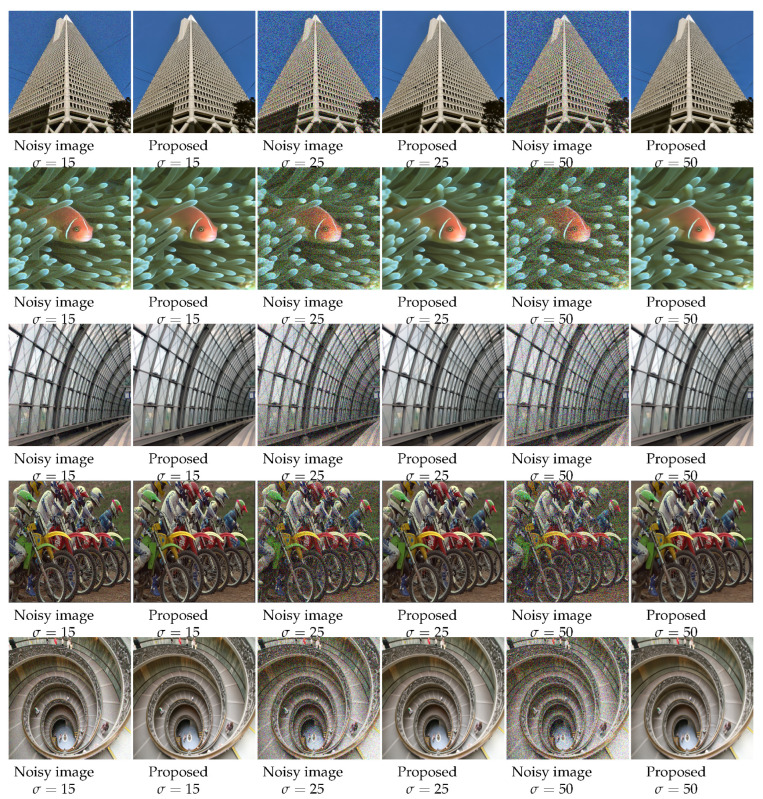
Sample results for different datasets for σ=15, 25, and 50 (for best view, zooming in is recommended).

**Figure 11 sensors-22-07080-f011:**
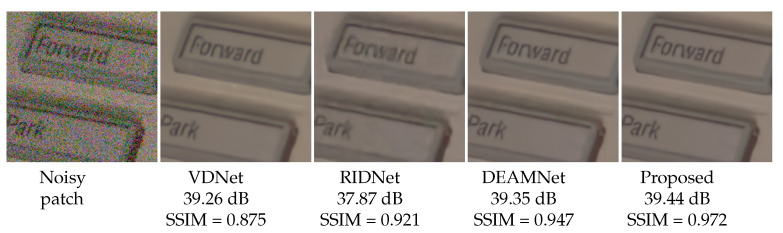
Visual quality comparison with PSNR and SSIM scores for the SIDD dataset with real noises (for best view, zooming in is recommended).

**Figure 12 sensors-22-07080-f012:**
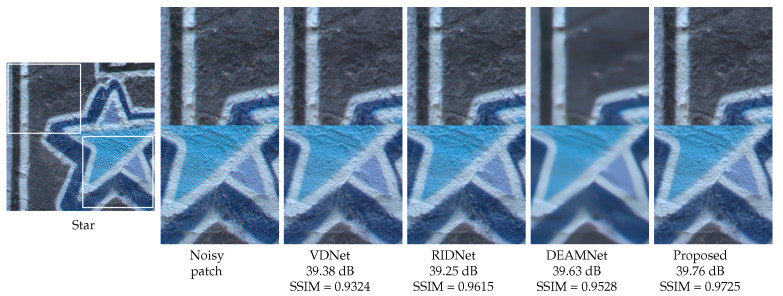
Visual quality comparison with PSNR and SSIM scores for “Star” from the DnD dataset with real noises (for best view, zooming in is recommended).

**Figure 13 sensors-22-07080-f013:**
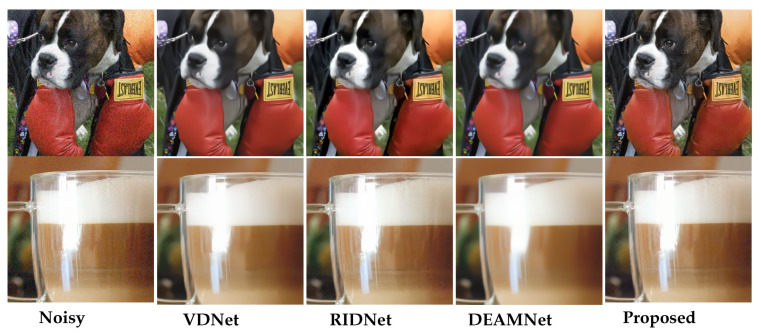
Visual quality comparison for “Dog" and “Glass” from the RN15 dataset. RN15 dataset is a set of real noise images without the clean image counterparts (for best view, zooming in is recommended).

**Figure 14 sensors-22-07080-f014:**
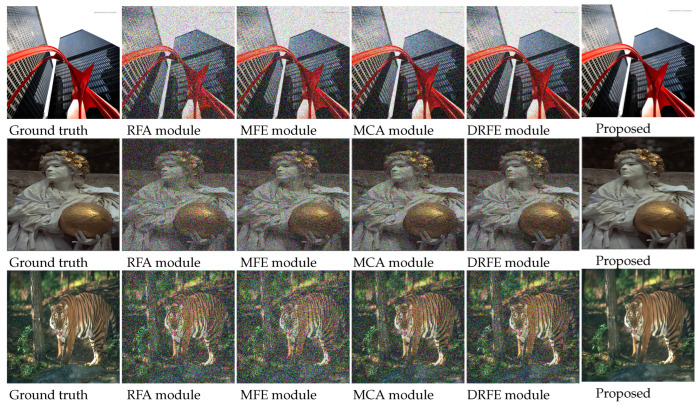
Sample results for all four modules separately (for best view, zooming in is recommended).

**Table 1 sensors-22-07080-t001:** Quantitative comparison results of the competing methods with AWGN noise levels σ=15, 25, 50 on kodak24, BSD68, and Urban100. Top results are in bold, and second-best results are underlined.

Method	Metrics		σ= 15			σ= 25			σ= 50	
		BSD68	Kodak24	Urban100	BSD68	Kodak24	Urban100	BSD68	Kodak24	Urban100
BM3D [2]	PSNR	32.37	31.07	32.35	29.97	28.57	29.70	26.72	25.62	25.95
	SSIM	0.8952	0.8717	0.9220	0.8504	0.8013	0.8777	0.7676	0.6864	0.7791
WNNM [12]	PSNR	32.70	31.37	32.97	30.28	28.83	30.39	27.05	25.87	26.83
	SSIM	0.8982	0.8766	0.9271	0.8577	0.8087	0.8885	0.7775	0.6982	0.8047
DnCNN [19]	PSNR	32.86	31.73	31.86	30.06	28.92	29.25	27.18	26.23	26.28
	SSIM	0.9031	0.8907	0.9255	0.8622	0.8278	0.8797	0.7829	0.7189	0.7874
FFDNet [28]	PSNR	32.75	31.63	32.43	30.43	29.19	29.92	27.32	26.29	26.28
	SSIM	0.9027	0.8902	0.9273	0.8634	0.8289	0.8886	0.7903	0.7245	0.8057
IrCNN [20] ${}^{1}$	PSNR	31.67	33.60	31.85	29.96	30.98	28.92	26.59	27.66	25.21
	SSIM	0.9318	0.9247	0.9493	0.8859	0.8799	0.9101	0.7899	0.7914	0.8168
ADNet [43]	PSNR	32.98	31.74	32.87	30.58	29.25	30.24	27.37	26.29	26.64
	SSIM	0.9050	0.8916	0.9308	0.8654	0.8294	0.8923	0.7908	0.7216	0.8073
RIDNet [5]	PSNR	32.91	31.81	33.11	30.60	29.34	30.49	27.43	26.40	26.73
	SSIM	0.9059	0.8934	0.9339	0.8672	0.8331	0.8975	0.7932	0.7267	0.8132
VDN [44]	PSNR	**33.90**	**34.81**	33.41	**31.35**	**32.38**	30.83	28.19	**29.19**	**28.43**
	SSIM	0.9243	0.9251	0.9339	0.8713	0.8842	0.8361	0.8014	0.7213	0.8212
DEAMNet [8]	PSNR	33.19	31.91	33.37	30.81	29.44	30.85	27.74	26.54	27.53
	SSIM	0.9097	0.8957	0.9372	0.8717	0.8373	0.9048	0.8057	0.7368	0.8373
Proposed	PSNR	33.85	32.90	**33.97**	31.32	30.67	**31.52**	**29.02**	28.12	28.25
	SSIM	**0.9603**	**0.9517**	**0.9621**	**0.9150**	**0.9246**	**0.9241**	**0.8831**	**0.8782**	**0.8755**

^1^ The PSNR results for Kodak24 and BSD68 were obtained from the IrCNN implementation from (https://github.com/cszn/IRCNN, accessed on 10 October 2021). We want to note that our results are significantly different from the results reported in [45].

**Table 2 sensors-22-07080-t002:** Real-image denoising results of several existing methods on SIDD and DnD dataset. Top results are in bold, and second best results are underlined.

Dataset	Metrics	BM3D	DnCNN	FFDNet	VDN	RIDNet	DEAMNet	Proposed
SIDD [42]	PSNR	25.65	23.66	29.30	39.26	37.87	39.35	**39.55**
	SSIM	0.685	0.583	0.694	0.944	0.943	0.955	**0.964**
DnD [41]	PSNR	34.51	32.43	37.61	39.38	39.25	39.63	**39.76**
	SSIM	0.8507	0.7900	0.9115	0.9518	0.9528	0.9531	**0.9617**

**Table 3 sensors-22-07080-t003:** Running time (in seconds) and parameter comparison.

Method	Size $256^{2}$	Size $512^{2}$	Size $1024^{2}$	Parameters
BM3D [2]	0.76	3.12	12.82	-
WNNM [12]	210.26	858.04	3603.68	-
DnCNN [19]	0.01	0.05	0.16	558 k
IrCNN [20]	0.012	0.038	0.146	-
FFDNet [28]	0.01	0.05	0.11	490 k
AINDNet [33]	0.05	0.03	0.80	13,764 k
ADNet [43]	0.02	0.06	0.20	519 k
VDN [44]	0.04	0.07	0.19	7817 k
RIDNet [5]	0.07	0.21	0.84	1499 k
DEAMNet [8]	0.05	0.19	0.73	2225 k
Proposed	0.031	0.11	0.42	846 k

**Table 4 sensors-22-07080-t004:** Removing different modules from the ensemble and comparing their results on different datasets.

Dataset	PSNR and SSIM	RFA Module Removed	MFE Module Removed	MCA Module Removed	DRFE Module Removed
BSD68	33.85	30.65	28.66	29.30	30.26
	0.9603	0.885	0.783	0.824	0.855
Kodak24	32.90	29.51	27.43	28.61	30.38
	0.9517	0.7507	0.6900	0.8115	0.8518
Urban100	33.97	30.48	27.75	30.54	31.38
	0.9621	0.7824	0.6192	0.7822	0.8766

## Data Availability

The datasets that support the findings of this study are openly available in BSD at https://www2.eecs.berkeley.edu/Research/Projects/CS/vision/grouping/segbench/, Kodak at http://r0k.us/graphics/kodak/, Urban at https://github.com/majedelhelou/denoising_datasets/tree/main/CUrban100/, DND at https://noise.visinf.tu-darmstadt.de/, SIDD at https://www.eecs.yorku.ca/~kamel/sidd/benchmark.php, and RN15 at https://demo.ipol.im/demo/125/archive/.

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
