# Peer review of "Denoising Single Images by Feature Ensemble Revisited"

_sensors, 2022, doi:10.3390/s22187080_

Round 1

Reviewer 1 Report

1.     From Fig. 8 & 9, VDN [43] outperforms the proposed one, which is also evident in table 1. Hence, justify how the proposed method is better than all other methods.

2.     Provide sample results of σ = 15 & 25 for the clarity of performance analysis.

3.     Time complexity analysis may vary with different machine setups. Hence, this is not a significant factor in the performance analysis.

4.     Fig. 2 requires appropriate label representation, as mentioned in the explanation of it.

5.     The authors should have provided the details about different synthetic types of noise images used and the proposed method's performance variance of these noises.

Author Response

We thank the reviewers for detail comments and questions.

Please see the attachment for all the point-by-point responses.

Ho Yub jung

Reviewer 2 Report

1. My major concerns is that the technical novelty should be clarified  and the motivation is not clear. Please specify the importance of the proposed solution.

2. In the conclusion section, the limitations of the proposed method must be discussed by the authors and to related need of further work

3. Overall contribution of proposed work was well presented and Experimental result shows the accuracy results of the proposed methodology.

Author Response

(The authors gave the same response as above.)

Reviewer 3 Report

This paper presents a denoising network by feature decomposition and recombination with four modules.

The idea is similar to conventional dual-domain denoising. It is interesting and has been verified in experiments. However, I still have some concerns:

1. Although the method performed well in the experiment, its motivation needs to be clarified. Why is it better than existing methods? Please analyze it.

2. In Section 2, please analyze the advantages of the proposed method compared with the recent deep learning-based denoising methods.

3.Although the authors have given the specific details of each module, i still have some questions: what are the effects of these four modules? what are their connections? Are they complementary? if yes, why?

4. Ablation analysis needs to be added in the experiments to verify the role of each module and the influence of parameters.

5. I have doubts about the data in Table 1. The original IrCNN[20] does not have the networks with noise intensities of 15 and 25. Does the authors retrain the networks? In addition, we all know that IrCNN has a very compact network structure and relatively poor performance. But in the data given by the authors, it often gets the second-best results.

6. What puzzles me most is the results of Figure 8. The PSNR value of the proposed method ranked second, but the visual effect was far better than that of the VDN method ranked first. Their PSNR difference is as much as 1dB. 

In addition, the PSNR value of IrCNN ranked third, but the visual effect was far worse than that of ADNET. Their PSNR difference is also as much as 1dB. 

The authors need to provide evidence to prove the authenticity of the data.

Author Response

We thank the reviewers for detail comments and questions.

Please see the Report Notes for all the point-by-point responses.

Ho Yub jung

Round 2

Reviewer 1 Report

The authors carried out appropriate responses to all the comments.

Author Response

We are glad we were able to address all the concerns.

Reviewer 3 Report

Although the revised version has been improved, I strongly recommend that the authors recheck the experimental data to ensure the authenticity of the data.

The review found that many data were wrong or untrue, such as:

1. In Table 1, the PSNR result of the VDN method with AWGN noise level σ = 25 on BSD68 is 31.25 dB, while it is repored in [43] (see Table 2) that it should be 31.35 dBThe re-run error will not be as much as 0.1dB.

2. It is repored in [01] (see Table 2) that the average PSNR(dB) result of the IRCNN method for noise level  50 on Kodak24 is 28.93 dB, while the one in the revised version is 27.66 dB. The reviewer has rechecked in person, and the result is that the former is correct.

[01] Zhang K ,  Li Y ,  Zuo W , et al. Plug-and-Play Image Restoration with Deep Denoiser Prior. IEEE Transactions on Pattern Analysis and Machine Intelligence, 2021.

3. The reviewer has also rechecked the result in Figure 8 which is also from Kodak24. The PSNR of IRCNN is 26.88 dB, while the SSIM is 0.7415, which are inconsistent with the modified results.

4. Some data are unreasonable. For example, in Table1, the PSNR and SSIM results of the VDN method with AWGN noise level σ = 50 on Kodak24 are 29.19 dB and 0.7213, while that of the proposed method are 28.12 dB and 0.8782. It is unreasonable that the trends of PSNR and SSIM are so different. If the authors insist that this is reasonable, please provide the denoised visual results of these two algorithms for review.

Author Response

We appreciate the constructive comments and suggestions provided by the reviewer. The reviewer raised some very important and valid points. We have faithfully address the reviews with best of our knowledge. The point by point responses are presented in pdf.
